ecology, environmental science

pharmaceutical pollution, thermal stress,
multi-stressor, fluoxetine, life history,
*Daphnia magna*

**Author for correspondence:**
Lucinda C. Aulsebrook
e-mail: lucinda.aulsebrook@monash.edu

Electronic supplementary material is available
online at https://doi.org/10.6084/m9.figshare.
c.5821254.

# Warmer temperatures limit the effects of antidepressant pollution on life-history traits

Lucinda C. Aulsebrook, Bob B. M. Wong and Matthew D. Hall

School of Biological Sciences, Monash University, Melbourne, VIC 3800, Australia

LCA, 0000-0002-8210-2400; BBMW, 0000-0001-9352-6500; MDH, 0000-0002-4738-203X

Pharmaceutical pollutants pose a threat to aquatic ecosystems worldwide. Yet, few studies have considered the interaction between pharmaceuticals and other chronic stressors contemporaneously, even though the environmental challenges confronting animals in the wild seldom, if ever, occur in isolation. Thermal stress is one such environmental challenge that may modify the threat of pharmaceutical pollutants. Accordingly, we investigated how fluoxetine (Prozac), a common psychotherapeutic and widespread pollutant, interacts with temperature to affect life-history traits in the water flea, *Daphnia magna*. We chronically exposed two genotypes of *Daphnia* to two ecological relevant concentrations of fluoxetine (30 ng l$^{-1}$ and 300 ng l$^{-1}$) and a concentration representing levels used in acute toxicity tests (3000 ng l$^{-1}$) and quantified the change in phenotypic trajectories at two temperatures (20°C and 25°C). Across multiple life-history traits, we found that fluoxetine exposure impacted the fecundity, body size and intrinsic growth rate of *Daphnia* in a non-monotonic manner at 20°C, and often in genotypic-specific ways. At 25°C, however, the life-history phenotypes of individuals converged under the widely varying levels of fluoxetine, irrespective of genotype. Our study underscores the importance of considering the complexity of interactions that can occur in the wild when assessing the effects of chemical pollutants on life-history traits.

## 1. Introduction

Pharmaceutical pollution has become an increasing threat to ecosystems worldwide. Thousands of pharmaceuticals are used for medical and veterinary healthcare, and more than 600 have been detected in the environment [1]. A large number of these remain bioactive when excreted [2], and sewage treatment is often insufficient at removing these products [3]. Once in the environment, pharmaceuticals are often resistant to degradation [2] and have the ability to bioaccumulate and transfer through food webs [4]. As many pharmaceuticals are designed to elicit responses at low doses [5], and often target receptors that are evolutionarily conserved in a range of species [6], there are mounting concerns about how these drugs may be affecting populations in the wild [7,8].

One such pharmaceutical of concern is fluoxetine, the active ingredient of Prozac—one of the most commonly prescribed antidepressants in the world [9]. Fluoxetine has been found in aquatic systems globally [4,9,10], with levels as high as 500 ng l$^{-1}$ downstream from wastewater treatment plants [11]. Fluoxetine functions as an antidepressant by increasing the effect of the neurotransmitter serotonin by inhibiting transport proteins that reuptake serotonin [12]. Antidepressants that function in such a manner are classed as selective serotonin reuptake inhibitors (SSRIs) [12,13]. The target molecule of fluoxetine (serotonin transporter, 5-HTT) is present in a wide variety of taxa [14], which gives fluoxetine the potential to affect non-target organisms. In

fact, studies have shown that fluoxetine exposure can cause adverse effects in a range of aquatic organisms, including reduced fecundity in snails [15], impaired development in tadpoles [16] and disturbed behaviour in fish [17–19]. Few studies, however, have examined the impacts on life-history traits at field-relevant concentrations, which is surprising considering that changes in these traits are expected to have dire fitness and evolutionary consequences [7,20,21].

While many studies have shown that pharmaceuticals can affect wildlife, these typically investigate the direct effects of single toxicants on organisms and do not consider how pharmaceutical pollutants can interact with other stressors [22]. This limits our understanding of the ecological impacts of these pollutants, as in natural environments, organisms are exposed to a variety of biotic and abiotic stressors simultaneously, and the combined effects of stressors are rarely additive [23,24]. Instead, stressors commonly interact to produce effects markedly different from the sum of each isolated effect, which can cause the true ecological impact of stressors to be misinterpreted [25,26]. For example, synergistic interactions between chemical pollutants and secondary stressors are frequently reported [27–30], perhaps due to the energetic cost of detoxification resulting in wildlife more vulnerable to secondary stressors. Conversely, antagonistic interactions could lessen the damage caused by pharmaceuticals, due to one stressor enabling tolerance to another, or having opposing effects [31]. The severity of the impact of a pharmaceutical pollutant on an ecosystem may, therefore, depend on what other stressors are present in the environment, a notion overlooked by the majority of studies.

One important stressor to consider is temperature, due to its direct effects on growth, reproduction and development of organisms, particularly ectotherms [32]. Furthermore, global change is causing temperature to become an increasingly concerning stressor to many ecosystems [33]. Increased temperature is often shown to exacerbate the effects of pollutants [28,34,35], via increasing toxicity [36] or reproductive costs [37]. While less common, there are also cases of heat stress reducing the effects of pollutants [28,38,39], through temperature and pollutants having opposing effects [40], or temperature disrupting the mechanism responsible for the pollutant's effects [41]. Nevertheless, few studies have investigated interactions between increased temperature and exposure to psychoactive pharmaceutical pollutants such as fluoxetine.

Another limitation of the majority of studies on pharmaceutical pollution is that they typically examine the effects on individual traits only [5,15]. While these studies provide important insights into how organisms may be impacted by pollutants, they can lead to contrasting or confusing results because the changes in phenotype may vary in magnitude or direction depending on the specific trait being measured [42]. In reality, organism phenotypes are comprised numerous traits, many covarying and capturing the full extent of environmental interactions requires understanding shifts in these integrated phenotypes [43,44]. Phenotypic trajectory analysis can achieve this through exploring phenotype shifts across multivariate space, allowing a more holistic understanding of how stressors may affect a population [45,46].

Here, we examined the consequences of fluoxetine exposure on life-history trait phenotypes under differing temperature treatments using *Daphnia magna*, a commonly used model organism in ecotoxicology [47]. The temperature treatments used were 20°C, which is the standard cultivation temperature of *Daphnia* [26,48–50], and 25°C, which is known to affect 'pace-of-life' traits, often leading to significantly faster maturation, earlier offspring release and smaller size at maturity, but at the expense of reduced survival and lifetime fecundity (e.g. [51–53]). For our fluoxetine exposure treatments, we compared a freshwater control with two environmentally realistic concentrations: 30 ng l$^{-1}$, representing levels commonly detected at surface waters and 300 ng l$^{-1}$, which represents approximate concentrations detected at wastewater outlets [54]. In addition, we included an extreme concentration of 3000 ng l$^{-1}$ as a comparison to levels commonly used in acute toxicity tests (e.g. [55–57]), as well as a freshwater control with no fluoxetine. Using a variety of life-history traits, we then employed phenotypic trajectory analysis to compare the magnitude and direction of temperature-induced phenotypic shifts for *Daphnia* under varying fluoxetine exposures, allowing us to assess whether increased temperature might intensify or reduce the effects of pharmaceutical pollutants on wildlife.

## 2. Methods

### (a) Study system

*Daphnia magna* is a freshwater filter-feeding crustacean native to Eurasia. The species is a model organism in both evolutionary biology and aquatic toxicology as it reproduces rapidly, is sensitive to their chemical environment and plays an essential role in freshwater ecosystems as primary consumers [48]. *Daphnia magna* most frequently produce asexually via cyclic parthenogenesis [48], resulting in genetic clones, which allow stocks of single genotypes to be easily maintained in a laboratory environment. For the current study, we used two *Daphnia* genotypes derived from single clones: HU-HO-2 (herein HO2) from Hungary and BE-OHZ-M10 (herein M10) from Belgium. These geographically diverse clones are known to vary in several life-history traits (e.g. [49,50,58]), and allow us to investigate whether fluoxetine and temperature effects are likely to be genotype specific.

Prior to the experiment, three generations of *Daphnia* were housed individually in 70-ml jars filled with 45 ml of artificial *Daphnia* media [59,60]. The medium was replaced twice a week and each jar was fed daily with an ad libitum amount of algae (*Scenedesmus* spp.). Food levels were gradually increased in accordance to the needs of the animals, from 0.5 million cells per animal on day 1, to 5 million cells per animal from day 8 onwards. All animals were kept in incubators with an 18 : 6 h light–dark cycle at a fixed temperature of 20°C. Experimental animals were taken from clutch 3–4 of 126 parental *Daphnia* of each genotype. These were maintained under the same standard conditions as parental lines, with the exception of temperature, which was fixed at either 20°C or 25°C depending on treatment group.

### (b) Fluoxetine and temperature exposure

We used a factorial experimental design where the two *Daphnia* genotypes (M10 and HO2) were exposed to the two temperature treatments (20°C or 25°C), under the four different nominal fluoxetine concentrations (0 ng l$^{-1}$, 30 ng l$^{-1}$, 300 ng l$^{-1}$ and 3000 ng l$^{-1}$). Twenty individuals were used for each genotype–temperature–fluoxetine treatment combination. Fluoxetine treatments were produced by dissolving the desired amount of fluoxetine hydrochloride in small volume of methanol, as per previously established protocols [18,61,62], then dosing this

methanol into media before distributing the media across jars. The media for the control treatment was dosed with a similar volume of methanol, but with no fluoxetine hydrochloride. All animals were exposed to fluoxetine treatments at 1 day old, and fluoxetine dosing occurred at each water change (i.e. twice weekly).

At each water change, samples were drawn from fluoxetine-dosed media to monitor fluoxetine concentrations, with the measured effective concentrations after the exposure period in line with the initial nominal fluoxetine concentrations doses ($25.87 \pm 2.47$ ng $l^{-1}$, $197.5 \pm 10.31$ ng $l^{-1}$ and $1900 \pm 184.39$ ng $l^{-1}$, see electronic supplementary material). Water analysis was performed by Envirolab Services (MPL Laboratories; NATA accreditation: 2901; accredited for compliance with ISO/IEC: 17025) using gas chromatography-tandem mass spectrometry (7000C Triple Quadrupole GC-MS/MS, Agilent Technologies, Delaware, USA) following methods described in [62].

Individuals were monitored daily for survival and the number of offspring and clutches produced was counted twice a week at each water change. Fecundity was calculated as the total number of offspring produced by each individual during the course of the experiment. The experiment was terminated at 30 days, whereupon the body size of all remaining *Daphnia* was measured as the length of *Daphnia* from the top of the head above the eye to the base of the tail spine. Intrinsic rates of increase per individual ($r$) were calculated using the timing and number of offspring and then solving the Euler–Lotka equation (following [63]).

## (c) Statistical analysis

Statistical tests were conducted using R software v. 4.0.3 software [64]. We first implemented linear mixed-effect models for each of the life-history traits measured, using fluoxetine treatment, temperature treatment, genotype and interactive terms as fixed effect factors, and blocks as a random effect. Across all traits, we then performed a phenotypic trajectory analysis (PTA) in order to determine how temperate and fluoxetine interact to shape a life-history phenotype [44]. This approach quantifies the relative magnitude ($D$) and angle ($\theta$) of any shift in multi-variate phenotype (phenotypic trajectory) across temperature for each fluoxetine concentrations using a permutation-based MANOVA. Multivariate analyses were conducted using the RRPP package [65], traits were scaled to a mean of 0 and standard deviation of 1, and subsequently visualized using principal component analysis (PCA).

# 3. Results

## (a) Effects of fluoxetine vary by trait, genotype and temperature

We found that responses were often specific to the trait measured, concentration of fluoxetine, temperature and *Daphnia* genotype. The simplest effects were observed for the timing and size of first clutch, whereby fluoxetine had no significance effect on trait values, while the influence of temperature was much stronger for genotype HO2, accounting for the genotype by temperature interaction (table 1 and figure 1a,b). For all other traits, the influence of fluoxetine depended either on temperature (temperature × fluoxetine interaction, table 1) or the interplay between both temperature and genotype (three-way interaction, table 1). At lower temperatures, we typically observed far greater differences among the fluoxetine treatments than at higher temperatures.

At 20°C, we saw that HO2 *Daphnia* exposed to the two higher fluoxetine concentrations had higher fecundity and intrinsic growth. For M10, we observed a non-monotonic response whereby individuals exposed to the lowest fluoxetine concentration had suppressed fecundity and body size relative to the controls, while the highest fluoxetine concentration greatly increased these traits. By contrast, at 25°C we found that for both genotypes there were no significant differences in fecundity, body size and intrinsic growth across the different fluoxetine treatments (figure 1c,d,e).

## (b) Multivariate analysis indicates an antagonistic interaction between temperature and fluoxetine exposure for both genotypes

Across the different life-history traits, we observed a variety of responses to genotype, fluoxetine and temperature. After accounting for correlations among trait responses via a phenotypic trajectory analysis, however, we observed an overarching antagonistic interaction between temperature and fluoxetine concentration. We found no significant difference in the magnitude of the phenotype trajectories between *Daphnia* exposed to different fluoxetine treatments at 20°C and 25°C ($D$ in table 2), indicating that the strength of temperature-driven phenotypic change is not altered by fluoxetine exposure, at both ecologically relevant and extreme concentrations. Instead, there were significant differences in the angles of the phenotype trajectories for each fluoxetine treatment ($\theta$ in table 2), particularly for the M10 genotype of *Daphnia* whereby most angles were greater than 30°. This indicated a reduction of phenotype differences at 25°C compared to 20°C.

Visualization of the phenotypic trajectory analysis revealed that increases in temperature lead to a convergence of life-history phenotypes and a reduction of phenotype differences at 25°C compared to 20°C (figure 2). The phenotype trajectories of each fluoxetine treatment diverged along the PC2 axis at 20°C, which primarily accounts for variation in total offspring and intrinsic growth for HO2 (figure 2c), and total offspring and body size for M10 (figure 2d). These trajectories then converged at 25°C (figure 2a,b), with increased temperature described by large shifts across the PC1 axis (figure 2a,b), which, for HO2, is primarily driven by differences in size and timing of the first clutch (figure 2c), and, for M10, intrinsic growth as well as timing of the first clutch (figure 2d).

# 4. Discussion

We found that even trace amounts of fluoxetine can affect a variety of life-history traits in *Daphnia*. While previous studies have indicated that fluoxetine exposure can alter fecundity in *Daphnia* [47,57] as well as other invertebrates [66,67], effects at exposure concentrations lower than 10 µg have rarely been seen before. We observed that ecologically important traits such as fecundity, body size and intrinsic growth were all affected by the lowest concentration of fluoxetine (30 ng $l^{-1}$). In specific cases (*Daphnia* genotype M10 for example), fluoxetine even had a non-monotonic effect on fecundity and body size, whereby the lowest concentration induced the greatest phenotypic change. Non-monotonic responses are increasingly being reported in studies

**Table 1.** Effects of genotype, temperature and fluoxetine treatment, as well as interactions between these terms, on total offspring, bodysize, age at first clutch, size of first clutch and intrinsic growth of *Daphnia magna*. Analysis was performed using linear mixed effect models on each trait.

| trait | term | $\chi^2$ | d.f. | *p*-value |
|---|---|---|---|---|
| size of first clutch | genotype | 3.049 | 1 | 0.081 |
| | temperature | 88.942 | 1 | *<0.001* |
| | fluoxetine treatment | 2.772 | 3 | 0.428 |
| | genotype : temperature | 49.495 | 1 | *<0.001* |
| | genotype : fluoxetine treatment | 2.701 | 3 | 0.440 |
| | temperature : fluoxetine treatment | 0.749 | 3 | 0.862 |
| | genotype : temperature : fluoxetine treatment | 6.164 | 3 | 0.104 |
| age at first clutch | genotype | 382.985 | 1 | *<0.001* |
| | temperature | 129.106 | 1 | *<0.001* |
| | fluoxetine treatment | 5.174 | 3 | 0.160 |
| | genotype : temperature | 165.404 | 1 | *<0.001* |
| | genotype : fluoxetine treatment | 1.146 | 3 | 0.766 |
| | temperature : fluoxetine treatment | 3.130 | 3 | 0.372 |
| | genotype : temperature : fluoxetine treatment | 2.217 | 3 | 0.529 |
| total offspring | genotype | 315.303 | 1 | *<0.001* |
| | temperature | 14.128 | 1 | *<0.001* |
| | fluoxetine treatment | 15.317 | 3 | *0.002* |
| | genotype : temperature | 17.900 | 1 | *<0.001* |
| | genotype : fluoxetine treatment | 0.232 | 3 | 0.972 |
| | temperature : fluoxetine treatment | 17.062 | 3 | *0.001* |
| | genotype : temperature : fluoxetine treatment | 9.550 | 3 | *0.023* |
| intrinsic growth (r) | genotype | 807.281 | 1 | *<0.001* |
| | temperature | 0.735 | 1 | 0.391 |
| | fluoxetine treatment | 9.629 | 3 | *0.022* |
| | genotype : temperature | 118.680 | 1 | *<0.001* |
| | genotype : fluoxetine treatment | 3.137 | 3 | 0.371 |
| | temperature : fluoxetine treatment | 11.400 | 3 | *0.010* |
| | genotype : temperature : fluoxetine treatment | 6.822 | 3 | 0.078 |
| body size | genotype | 278.676 | 1 | *<0.001* |
| | temperature | 0.000 | 1 | 0.986 |
| | fluoxetine treatment | 7.835 | 3 | 0.050 |
| | genotype : temperature | 41.916 | 1 | *<0.001* |
| | genotype : fluoxetine treatment | 5.709 | 3 | 0.127 |
| | temperature : fluoxetine treatment | 10.987 | 3 | *0.012* |
| | genotype : temperature : fluoxetine treatment | 1.879 | 3 | 0.598 |

investigating the effects of fluoxetine on wildlife (e.g. [68–71]), possibly because, at higher concentrations, receptors become desensitized, or negative feedback loops are induced [72]. Our results highlight the need to use environmentally realistic concentrations when investigating the effects of pharmaceuticals on ecosystems, as effects at these concentrations may be notably different and sometimes more severe than effects seen at the higher concentrations typical of many studies.

The introduction of another ecological challenge, thermal stress, fundamentally altered the phenotypic consequences of fluoxetine exposure. A 5°C increase in temperature led to all treatments, freshwater control and fluoxetine exposures alike, converging on a common life-history phenotype (figure 2), suggesting that rising temperatures may potentially reduce the net phenotypic effects of fluoxetine pollution on an individual's life-history in some contexts. One potential explanation for this process is that fluoxetine is eliminated more rapidly as temperature increases [73,74], reducing its impact. Increases in temperate are known to accelerate the pace of life for an organism, favouring higher reproductive turn over and shorter lifespans [75], and in this case perhaps the rapid elimination of fluoxetine. It is also possible that increased temperature may disrupt the mechanism via

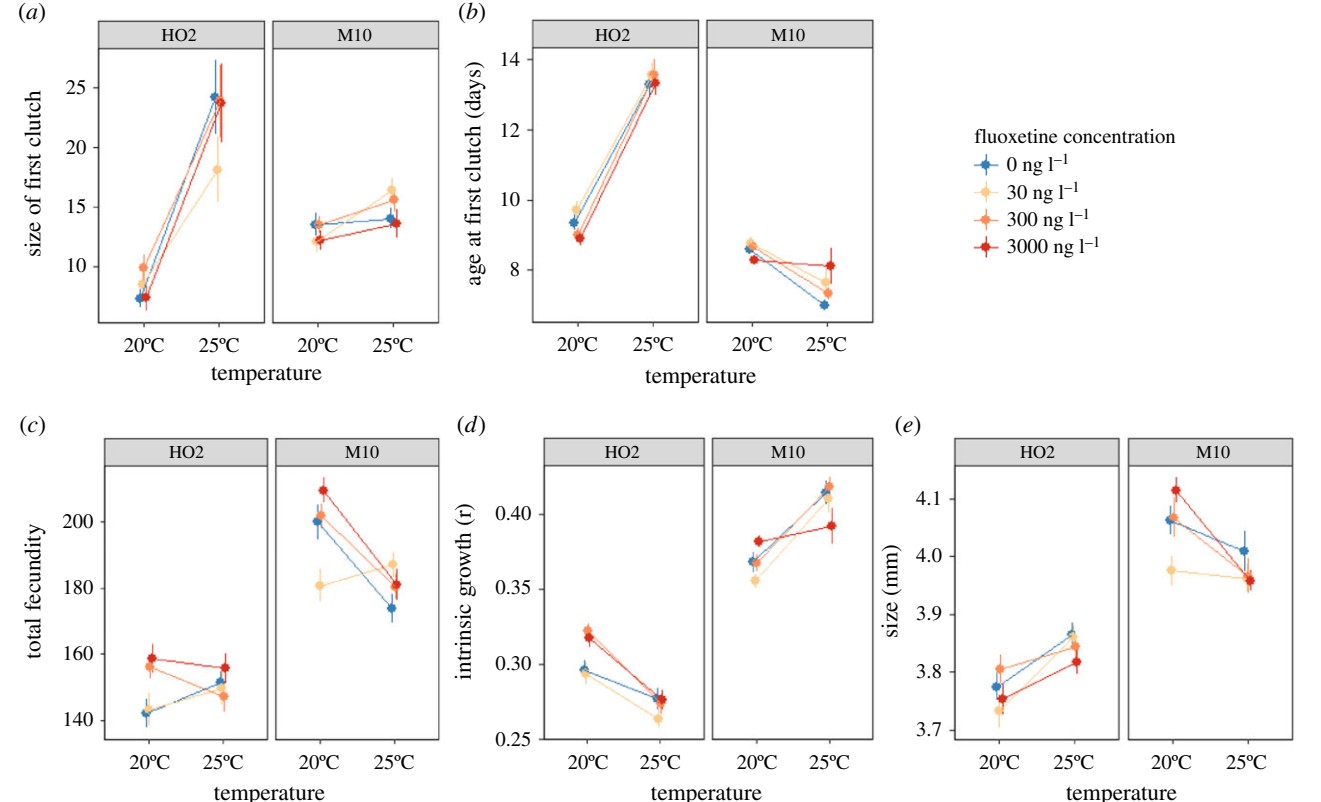

**Figure 1.** Univariate responses of two different genotypes (HO2 and M10) of *Daphnia magna* exposed to four different fluoxetine treatments (0 ng l$^{-1}$, 30 ng l$^{-1}$, 300 ng l$^{-1}$ and 3000 ng l$^{-1}$) at two different temperatures (20°C to 25°C). Means with standard error bars are depicted. (Online version in colour.)

**Table 2.** Phenotypic trajectory analysis (PTA) showing differences in magnitude (*D*) and angle (*θ*) in temperature driven phenotype shifts across each fluoxetine treatment comparison within each genotype. Phenotypes are based on five life-history traits.

| genotype | treatment comparisons | magnitude difference (*D*) | *Z* | *p*-value | angle difference (*θ*) | *Z* | *p*-value |
|---|---|---|---|---|---|---|---|
| HO2 | 0 ng l$^{-1}$ : 30 ng l$^{-1}$ | 0.079 | −0.891 | 0.792 | 18.859 | 0.378 | 0.322 |
| | 0 ng l$^{-1}$ : 300 ng l$^{-1}$ | 0.280 | 0.128 | 0.387 | 31.642 | 2.128 | *0.034* |
| | 0 ng l$^{-1}$ : 3000 ng l$^{-1}$ | 0.227 | −0.172 | 0.482 | 21.349 | 0.679 | 0.235 |
| | 30 ng l$^{-1}$ : 300 ng l$^{-1}$ | 0.359 | 0.508 | 0.263 | 30.102 | 1.909 | *0.044* |
| | 30 ng l$^{-1}$ : 3000 ng l$^{-1}$ | 0.307 | 0.202 | 0.372 | 22.757 | 0.885 | 0.188 |
| | 300 ng l$^{-1}$ : 3000 ng l$^{-1}$ | 0.053 | −1.057 | 0.882 | 10.540 | −0.844 | 0.787 |
| M10 | 0 ng l$^{-1}$ : 30 ng l$^{-1}$ | 0.067 | −1.019 | 0.851 | 41.647 | 2.659 | *0.011* |
| | 0 ng l$^{-1}$ : 300 ng l$^{-1}$ | 0.061 | −1.107 | 0.880 | 15.653 | −0.491 | 0.659 |
| | 0 ng l$^{-1}$ : 3000 ng l$^{-1}$ | 0.467 | 0.511 | 0.267 | 39.854 | 2.272 | *0.025* |
| | 30 ng l$^{-1}$ : 300 ng l$^{-1}$ | 0.129 | −0.771 | 0.747 | 36.260 | 1.964 | *0.038* |
| | 30 ng l$^{-1}$ : 3000 ng l$^{-1}$ | 0.534 | 0.916 | 0.181 | 64.922 | 5.325 | *0.001* |
| | 300 ng l$^{-1}$ : 3000 ng l$^{-1}$ | 0.406 | 0.373 | 0.322 | 32.108 | 1.370 | 0.104 |

which fluoxetine induces a response, as has been seen in other examples of antagonistic interactions between temperature and chemical pollutants [41,76]. Regardless of the underlying mechanism, our results suggest that the effects of the pollutant fluoxetine will not necessarily be exacerbated under the rise in temperatures predicted for many scenarios of global change. This demonstrates that while climate change is often predicted to amplify threats to ecosystems, this is not always inevitable (see also [39,77]), although due

to the complex nature of ecosystems, the exact effects are likely to depend on context, such as the type of pollutant, the type of thermal change, and as we discuss below, the genetic background of the exposed individual. In particular, in any lake or pond where *Daphnia* exist, they will likely be exposed to a variety of temperatures, due to spatial and temporal variation in thermal regimes, and their own ability to migrate vertically in the water column [78–80]. In the wild, the potential for temperature change to limit the impact of fluoxetine will

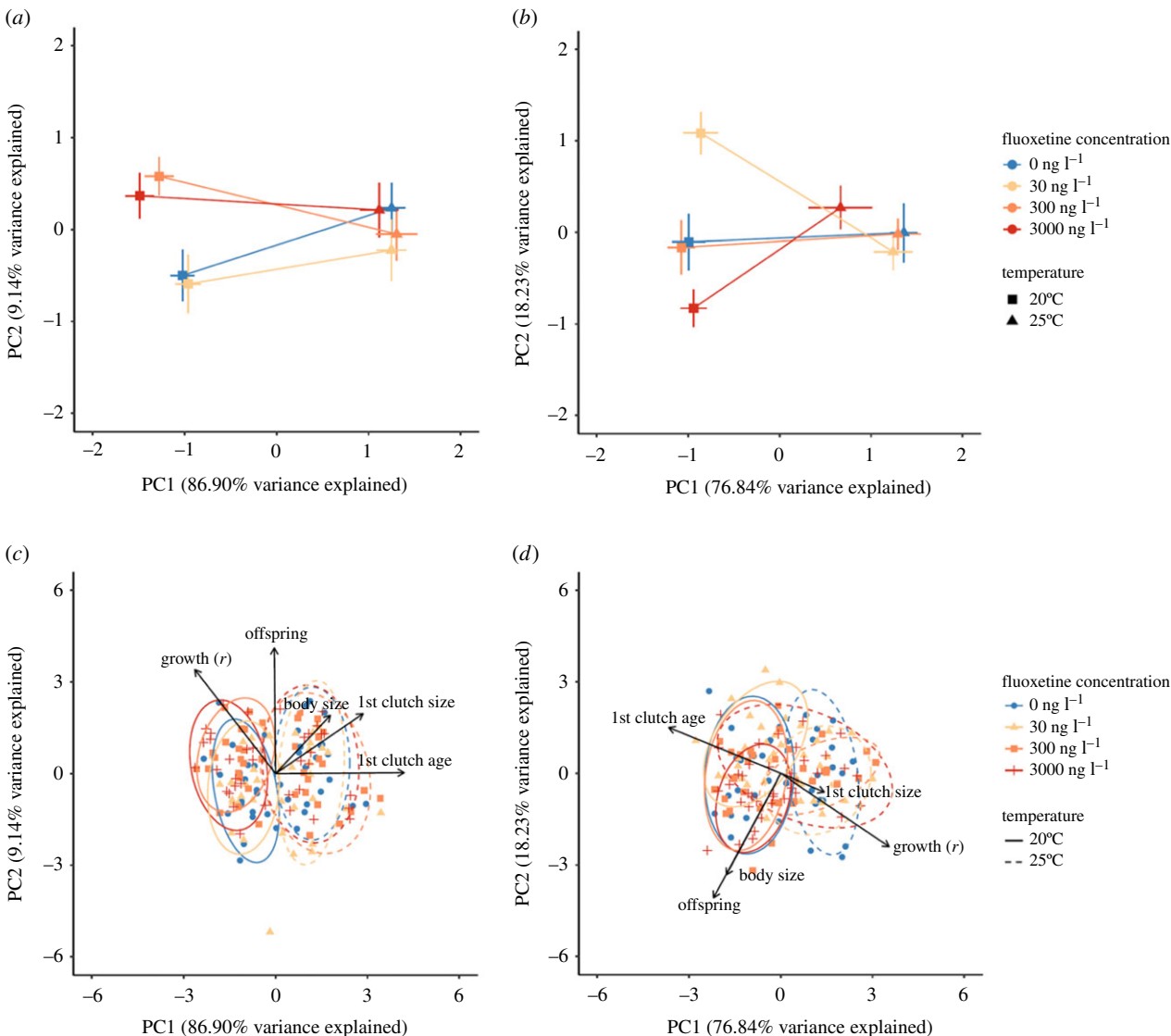

**Figure 2.** Principal component plots depicting: (*a*) phenotype trajectories of HO2 genotype *Daphnia magna* in response to temperature and fluoxetine treatments, (*b*) phenotype trajectories of M10 genotype *Daphnia*. (*c*) All observations for HO2 *Daphnia* grouped according to temperature and fluoxetine treatments and (*d*) all observations for M10 *Daphnia* grouped according to temperature and fluoxetine treatments. Contributions of each life-history trait (first clutch age, first clutch size, offspring, body size and intrinsic growth) toward the PC1 and PC2 axis for each genotype are shown (*c,d*). Ellipses represent 95% confidence bands. (Online version in colour.)

depend strongly on this fine-scale variation in temperature and the exposed individual's own thermal preference.

While very few studies have investigated interactions between temperature and fluoxetine specifically, the studies that exist have yielded conflicting results. Barbosa *et al.* [81], for example, found that fluoxetine exposure and increased variation in temperature had a synergistic effect on *Daphnia* lifetime reproductive success and population growth rate, while Wiles *et al.* [82] found no interactions between temperature stress and fluoxetine exposure on guppy behaviour. Given that, in our study, fluoxetine by temperature responses appeared to be trait-specific under the univariate analyses, we suggest that these contrasting results should be expected on a trait-by-trait basis. As our multivariate analysis revealed, it is only by integrating across many traits that a consensus may emerge for how interactions between fluoxetine and other ecological relevant stressors might influence phenotypic change in a population. Otherwise, viewing the effect of pollutants on single traits in isolation may fundamentally under or overestimate the consequences of pharmaceutical pollutants for natural populations.

We also found that the effects of fluoxetine and temperature were often affected by the genotype of *Daphnia*, a factor that has rarely been considered when investigating effects of pharmaceutical pollutants. More commonly the effects of toxicants are typically tested using only a single standard background genotype ([83–87] but see [52]), overlooking a considerable source of variation underlying a population's response to pharmaceuticals. While we only examined two genotypes, these have been shown to vary considerably in a variety of contexts [49,50,58], and, therefore, give an indication of the potential for genotype-specific responses. Employing a variety of genotypes will help to further explore how genetic variability could shape a population's net response to pharmaceutical pollution, as well as its potential to evolve in response to this source of human-induced environmental change.

Overall, our findings highlight the complexity of wildlife responses to chemical pollutants, where secondary factors such as temperature can fundamentally alter phenotypic consequences in unforeseen ways. Indeed, warmer temperatures appear to lessen the effects of fluoxetine on an organism's life history, suggesting that the effects of this widespread

pharmaceutical will not necessarily be made worse under common scenarios of global change. Such a result could easily have been overlooked if only a single host trait was measured, if host genotype had not been taken into account, or if ecologically relevant concentrations of the pollutant were not employed. Accordingly, to understand the full impact of pharmaceuticals on wildlife, we suggest that future studies capture more of the complexity of natural populations, where genetic variability, complex multivariate phenotypes, and the potential for non-monotonic responses, interact to shape individual performance or overall ecosystem function in the face of pharmaceutical pollutants.

Data accessibility. Data are available from the Dryad Digital Repository: https://doi.org/10.5061/dryad.8w9ghx3mh [88].

Authors' contributions. L.C.A.: conceptualization, formal analysis, investigation, methodology, visualization, writing—original draft, writing—review and editing; B.B.M.W.: conceptualization, methodology, resources, supervision, writing—review and editing; M.D.H.: conceptualization, formal analysis, methodology, resources, supervision, visualization, writing—review and editing.

All authors gave final approval for publication and agreed to be held accountable for the work performed therein.

Competing interests. We declare we have no competing interests.

Funding. This work was supported by the Australian Research Council (FT190100014 to B.B.M.W. and FT180100248 to M.D.H.) and an Australian Government Research Training Program Scholarship (to L.C.A.).

Acknowledgements. We thank Isobel Booksmythe for her assistance with laboratory work, David Williams and Envirolab Services for analytical testing of water samples, and Tobias Hector for assistance with analysis.

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
