## [Peer Review File · Proceedings of the Royal Society B: Biological Sciences]

Review History

RSPB-2021-2701.R0 (Original submission)

Review form: Reviewer 1

Recommendation

Accept with minor revision (please list in comments)

Scientific importance: Is the manuscript an original and important contribution to its field?

Excellent

General interest: Is the paper of sufficient general interest?

Excellent

Quality of the paper: Is the overall quality of the paper suitable?

Excellent

Is the length of the paper justified?

Yes

Should the paper be seen by a specialist statistical reviewer?

No

Do you have any concerns about statistical analyses in this paper? If so, please specify them explicitly in your report.

No

It is a condition of publication that authors make their supporting data, code and materials available - either as supplementary material or hosted in an external repository. Please rate, if applicable, the supporting data on the following criteria.

Is it accessible?

Yes

Is it clear?

Yes

Is it adequate?

Yes

Do you have any ethical concerns with this paper?

No

Comments to the Author

This report explores the effects of the antidepressant fluoxetine on the life history responses of two clones of *Daphnia magna* at two temperatures, with a fully crossed design. The main findings of the study were that the effects of fluoxetine were non-monotonic, clone and temperature dependent. Their results highlight the difficulties of extrapolating from laboratory studies to complex environments.

The implementation and analysis of the study were good, including accurately validating the extremely low fluoxetine concentrations and their results do demonstrate the complexity in understanding impacts of pollutants at field-level scales.

The following comment may assist in improving the paper:

L54-L68. In this paragraph the authors discuss the issue of multiple stressors. However, I am not convinced that 25°C is actually a stress for *D. magna*. In fact, one of their clones had a higher intrinsic rate of increase at 25°C. I think this section would be better directed at exploring temperature as a variable that can affect the effects of toxicants rather than as a stressor.

L110. What were the rearing conditions of the lines? How many parents were used to produce the test generation? What brood number were they from?

L116. It is more conventional to present algal concentrations as cells/mL of C/mL. Was this amount of food ad libitum? Particularly at the higher temperature was there any evidence of food limitation? What was the pH of the water? Did this vary with temperature?

L119. 70% humidity. I thought it would be 100% humidity in the water!

L122. I can't find mention anywhere of how many individuals/treatment?

L140. Were survival and fecundity recorded daily? If it was only twice a week, significant errors can occur in the estimation of life history parameters in such a rapidly breeding taxa.

L158. Please state survival in all treatments. And please include data on timing and brood size in each instar for each treatment, at least in an appendix. Were all offspring born healthy? Did offspring differ in size?

L163. Although not statistically significant, the low size of first clutch at 25°C is noticeable and worth commenting on.

Fig. 2. There is too much information in graphs C and D. Either find a way to simplify or delete. I don't think they add a lot anyway.

Review form: Reviewer 2

Recommendation

Major revision is needed (please make suggestions in comments)

Scientific importance: Is the manuscript an original and important contribution to its field?

Excellent

General interest: Is the paper of sufficient general interest?

Excellent

Quality of the paper: Is the overall quality of the paper suitable?

Excellent

Is the length of the paper justified?

Yes

Should the paper be seen by a specialist statistical reviewer?

No

Do you have any concerns about statistical analyses in this paper? If so, please specify them explicitly in your report.

No

It is a condition of publication that authors make their supporting data, code and materials available - either as supplementary material or hosted in an external repository. Please rate, if applicable, the supporting data on the following criteria.

Is it accessible?

Yes

Is it clear?

Yes

Is it adequate?

Yes

Do you have any ethical concerns with this paper?

No

Comments to the Author

This manuscript is a novel contribution to unraveling the complex responses of organisms to multiple potential stressors (e.g., antidepressants and temperature). The experiments and analyses are sound and the results are well supported and of interest. The manuscript was well written and the results were presented very clearly. In general, the manuscript provides a clear description of the need for this research and a good description of the research conducted. These types of experiments are imperative for understanding how multiple stressors affect organisms in nature, where multiple stressors exist.

My only comments for revision are to request that the authors provide a bit more detail about the study organism and its thermal optimum and how the temperature treatment compares to their thermal optimum. The authors indicate that the higher temperatures represent a stressor, but do not provide adequate information in the text to ascertain how sensitive *D. magna* is to a 20 vs. 25C thermal regime.

Along the same lines, in the discussion the authors make some statements which I think need to be refined given thermal regimes in lakes, stratification and the potential behavior of *Daphnia*.

The statement "Indeed, warmer temperatures appear to negate the effects of fluoxetine on an organism's life-history, suggesting that the effects of this widespread pharmaceutical will not necessarily be made worse under common scenarios of global change." And other similar statements in the discussion could be refined because in any given lake or pond where *Daphnia* exist, they will be exposed to a variety of temperatures. *Daphnia* are also known to migrate vertically in the water column as well (thereby experiencing different temperatures).

So, although the point is certainly useful that there is a strong potential for temperature to interact with contaminants like the one studied here, how this plays out in a given ecosystem will be very complex. I would recommend that the authors avoid the word "negate" and stress the interactions of stressors and their findings demonstrate that the effects of this pollutant are strongly dependent on the temperature during exposure, at least within the rather narrow range that has been examined.

Decision letter (RSPB-2021-2701.R0)

05-Jan-2022

Dear Ms Aulsebrook

I am pleased to inform you that your manuscript RSPB-2021-2701 entitled "Warmer temperatures limit the effects of antidepressant pollution on life-history traits" has been accepted for publication in Proceedings B.

The referee(s) have recommended publication, but also suggest some minor revisions to your manuscript. Therefore, I invite you to respond to the referee(s)' comments and revise your manuscript. Because the schedule for publication is very tight, it is a condition of publication that you submit the revised version of your manuscript within 7 days. If you do not think you will be able to meet this date please let us know.

- 1) A text file of the manuscript (doc, txt, rtf or tex), including the references, tables (including captions) and figure captions. Please remove any tracked changes from the text before submission. PDF files are not an accepted format for the "Main Document".

2) A separate electronic file of each figure (tiff, EPS or print-quality PDF preferred). The format should be produced directly from original creation package, or original software format. PowerPoint files are not accepted.

3) Electronic supplementary material: this should be contained in a separate file and where possible, all ESM should be combined into a single file. All supplementary materials accompanying an accepted article will be treated as in their final form. They will be published alongside the paper on the journal website and posted on the online figshare repository. Files on figshare will be made available approximately one week before the accompanying article so that the supplementary material can be attributed a unique DOI.

It is a condition of publication that data supporting your paper are made available either in the electronic supplementary material or through an appropriate repository. Please see our Data Sharing Policies <https://royalsociety.org/journals/authors/author-guidelines/#data>.

Sincerely,

Dr Locke Rowe

Associate Editor

Board Member: 1

Comments to Author:

I have now obtained two expert reviews of this paper; both referees agree that this study is very well-done and will make a strong contribution to the field - I completely agree. As you will see below, both reviewers make a number of (relatively minor but very helpful) suggestions for improvements. I call particular attention to the issues of (1) the natural thermal optimum of the study organism and whether 25C indeed represents a stressor or not - a point that needs clarification; (2) the discussion and interpretation of the interaction of fluoxetine and temperature which might be context-dependent; and, on a related issue, (3) the notion that in natural situations may be much more complex: likely only few lakes in nature will consistently experience 20 or 25C across time and space in a persistent manner, and *Daphnia* might avoid particular temperatures by vertical migration - in essence, a bit more ecological realism should be added to the discussion. For the detailed comments by the referees please see below. This paper should be acceptable pending some minor revisions and will make a strong and interesting contribution to the field.

Reviewer(s)' Comments to Author:

Referee: 1

Comments to the Author(s)

This report explores the effects of the antidepressant fluoxetine on the life history responses of two clones of *Daphnia magna* at two temperatures, with a fully crossed design. The main findings of the study were that the effects of fluoxetine were non-monotonic, clone and temperature dependent. Their results highlight the difficulties of extrapolating from laboratory studies to complex environments.

The implementation and analysis of the study were good, including accurately validating the extremely low fluoxetine concentrations and their results do demonstrate the complexity in understanding impacts of pollutants at field-level scales.

The following comment may assist in improving the paper:

L54-L68. In this paragraph the authors discuss the issue of multiple stressors. However, I am not convinced that 25°C is actually a stress for *D. magna*. In fact, one of their clones had a higher intrinsic rate of increase at 25°C. I think this section would be better directed at exploring temperature as a variable that can affect the effects of toxicants rather than as a stressor.

L110. What were the rearing conditions of the lines? How many parents were used to produce the test generation? What brood number were they from?

L116. It is more conventional to present algal concentrations as cells/mL of C/mL. Was this amount of food ad libitum? Particularly at the higher temperature was there any evidence of food limitation? What was the pH of the water? Did this vary with temperature?

L119. 70% humidity. I thought it would be 100% humidity in the water!

L122. I can't find mention anywhere of how many individuals/treatment?

L140. Were survival and fecundity recorded daily? If it was only twice a week, significant errors can occur in the estimation of life history parameters in such a rapidly breeding taxa.

L158. Please state survival in all treatments. And please include data on timing and brood size in each instar for each treatment, at least in an appendix. Were all offspring born healthy? Did offspring differ in size?

L163. Although not statistically significant, the low size of first clutch at 25°C is noticeable and worth commenting on.

Fig. 2. There is too much information in graphs C and D. Either find a way to simplify or delete. I don't think they add a lot anyway.

Referee: 2

Comments to the Author(s)

This manuscript is a novel contribution to unraveling the complex responses of organisms to multiple potential stressors (e.g., antidepressants and temperature). The experiments and analyses are sound and the results are well supported and of interest. The manuscript was well written and the results were presented very clearly. In general, the manuscript provides a clear description of the need for this research and a good description of the research conducted. These types of experiments are imperative for understanding how multiple stressors affect organisms in nature, where multiple stressors exist.

My only comments for revision are to request that the authors provide a bit more detail about the study organism and its thermal optimum and how the temperature treatment compares to their thermal optimum. The authors indicate that the higher temperatures represent a stressor, but do not provide adequate information in the text to ascertain how sensitive *D. magna* is to a 20 vs. 25C thermal regime.

Along the same lines, in the discussion the authors make some statements which I think need to be refined given thermal regimes in lakes, stratification and the potential behavior of *Daphnia*.

The statement "Indeed, warmer temperatures appear to negate the effects of fluoxetine on an organism's life-history, suggesting that the effects of this widespread pharmaceutical will not necessarily be made worse under common scenarios of global change." And other similar statements in the discussion could be refined because in any given lake or pond where *Daphnia* exist, they will be exposed to a variety of temperatures. *Daphnia* are also known to migrate vertically in the water column as well (thereby experiencing different temperatures).

So, although the point is certainly useful that there is a strong potential for temperature to interact with contaminants like the one studied here, how this plays out in a given ecosystem will be very complex. I would recommend that the authors avoid the word "negate" and stress the interactions of stressors and their findings demonstrate that the effects of this pollutant are strongly dependent on the temperature during exposure, at least within the rather narrow range that has been examined.

Author's Response to Decision Letter for (RSPB-2021-2701.R0)

See Appendix A.

Decision letter (RSPB-2021-2701.R1)

12-Jan-2022

Dear Ms Aulsebrook

I am pleased to inform you that your manuscript entitled "Warmer temperatures limit the effects of antidepressant pollution on life-history traits" has been accepted for publication in Proceedings B.

You can expect to receive a proof of your article from our Production office in due course, please check your spam filter if you do not receive it. PLEASE NOTE: you will be given the exact page

length of your paper which may be different from the estimation from Editorial and you may be asked to reduce your paper if it goes over the 10 page limit.

Your article has been estimated as being 8 pages long. Our Production Office will be able to confirm the exact length at proof stage.

Data Accessibility section

Open Access

Paper charges

Sincerely,

Proceedings B

Appendix A

Dear Dr Rowe,

We are delighted by the positive response to our submission and take the opportunity to thank you, the editorial board member, and reviewers for the constructive input. As you will see below, we have carefully considered the editor and reviewer comments in revising our manuscript. We hope that the paper is now acceptable for publication, but are, of course, happy to make further changes as required.

Yours sincerely,

Lucinda Aulsebrook, Bob Wong and Matthew Hall

BOARD MEMBER'S COMMENTS TO THE AUTHOR

BOARD MEMBER COMMENT: I have now obtained two expert reviews of this paper; both referees agree that this study is very well-done and will make a strong contribution to the field - I completely agree. As you will see below, both reviewers make a number of (relatively minor but very helpful) suggestions for improvements. I call particular attention to the issues of (1) the natural thermal optimum of the study organism and whether 25C indeed represents a stressor or not - a point that needs clarification; (2) the discussion and interpretation of the interaction of fluoxetine and temperature which might be context-dependent; and, on a related issue, (3) the notion that in natural situations may be much more complex: likely only few lakes in nature will consistently experience 20 or 25C across time and space in a persistent manner, and *Daphnia* might avoid particular temperatures by vertical migration - in essence, a bit more ecological realism should be added to the discussion. For the detailed comments by the referees please see below. This paper should be acceptable pending some minor revisions and will make a strong and interesting contribution to the field.

RESPONSE: We have attended to all of the three main points raised by the editorial board member. First, we can confirm that 25°C does, indeed, represent a stressor. We have now clarified this point and provided details on the established responses to 25°C in the manuscript. Second, we have now adjusted the discussion and interpretation, as requested. Lastly, on the point about ecological complexity, we agree with the reviewer and have adjusted the discussion to highlight this complexity. For details of how we have specifically addressed these (and other) reviewer comments, please see our responses to reviewers below.

REFEREE 1:

REFEREE COMMENT: This report explores the effects of the antidepressant fluoxetine on the life history responses of two clones of *Daphnia magna* at two temperatures, with a

fully crossed design. The main findings of the study were that the effects of fluoxetine were non-monotonic, clone and temperature dependent. Their results highlight the difficulties of extrapolating from laboratory studies to complex environments. The implementation and analysis of the study were good, including accurately validating the extremely low fluoxetine concentrations and their results do demonstrate the complexity in understanding impacts of pollutants at field-level scales.

RESPONSE: We thank the referee for their positive feedback.

REFeree COMMENT: The following comment may assist in improving the paper: L54-L68. In this paragraph the authors discuss the issue of multiple stressors. However, I am not convinced that 25°C is actually a stress for *D. magna*. In fact, one of their clones had a higher intrinsic rate of increase at 25°C. I think this section would be better directed at exploring temperature as a variable that can affect the effects of toxicants rather than as a stressor.

RESPONSE: The reviewer raises a good point in that we should have provided more clarity that 25°C is indeed a stressor in this scenario, due to the *Daphnia* being cultivated at 20°C for many generations, and the fact that it is known that temperature variation, including increases by 5°C, is an established stressor (Brans *et al.* 2018 *Functional Ecology*, Sadler *et al.* 2019 *Environmental Pollution*, Hector *et al.* 2021 *Biology Letters*). While there can be cases of higher intrinsic growth, this often comes at the cost of reduced survival and overall fecundity. In any case, in light of the reviewer's comments, we have amended the text, as follows:

"The temperature treatments used were 20°C, which is the standard cultivation temperature of Daphnia [26, 48-50], and 25°C, which is known to affect "pace-of-life" traits, often leading to significantly faster maturation, earlier offspring release and smaller size at maturity, but at the expense of reduced survival and lifetime fecundity (e.g. [51-53])." (lines 90-94)

For added clarity, we have also now mentioned that the parental generations of *Daphnia* used in the experiment were cultivated at 20°C. The relevant text reads, as follows:

"Prior to the experiment, three generations of Daphnia... were kept at incubators with an 1:6h light dark cycle at a fixed temperature of 20°C." (lines 116-122)

REFeree COMMENT: L110. What were the rearing conditions of the lines? How many parents were used to produce the test generation? What brood number were they from?

RESPONSE: Fair questions. We have now included this information as follows:

“Prior to the experiment, three generations of Daphnia were housed individually in 70-ml jars filled with 45 mL of artificial Daphnia media [59, 60]. The medium was replaced twice a week and each jar was fed daily with an ad libitum amount of algae (Scenedesmus spp.). Food levels were gradually increased in accordance to the needs of the animals, from 0.5 million cells per animal on day 1, to 5 million cells per animal from day 8 onwards. All animals were kept in incubators with an 18:6h light dark cycle at a fixed temperature of 20°C. Experimental animals were taken from clutch 3-4 of 126 parental Daphnia of each genotype. These were maintained under the same standard conditions as parental lines, with the exception of temperature, which was fixed at either 20°C or 25°C depending on treatment group.” (lines 116-125)

REFeree COMMENT: L116. It is more conventional to present algal concentrations as cells/mL of C/mL. Was this amount of food ad libitum? Particularly at the higher temperature was there any evidence of food limitation? What was the pH of the water? Did this vary with temperature?

RESPONSE: While algae feeding can be presented in cells/mL, it is not uncommon for papers to use cells per animal (e.g. Hector *et al* 2019 *Global Change Biology*, Hall & Ebert 2012 *Proceedings B*, Ben-Ami *et al* 2010 *The American Naturalist*), which is our preference. We did not measure the pH of the water as part of the experiment, so we do not have this information. The food we used was ad libitum for both temperatures, and was replenished daily. We have included this in the text as follows:

“The medium was replaced twice a week and each jar was fed daily with an ad libitum amount of algae (Scenedesmus spp.).” (lines 117-119)

REFeree COMMENT: L119. 70% humidity. I thought it would be 100% humidity in the water!

RESPONSE: The incubators themselves were set to 70% humidity to limit the rate of evaporation in the water filled jars housing the *Daphnia*. In hindsight, the incubator humidity is not very relevant, so to avoid confusion we have now removed this information. The sentence now reads as follows:

“All animals were kept incubators with an 18:6h light dark cycle at a fixed temperature of 20°C.” (lines 120-122)

REFeree COMMENT: L122. I can't find mention anywhere of how many individuals/treatment?

RESPONSE: Apologies. We have now added this detail as follows:

“Twenty individuals were used for each genotype-temperature-fluoxetine treatment combination.” (lines 129-130)

REFEREE COMMENT: L140. Were survival and fecundity recorded daily? If it was only twice a week, significant errors can occur in the estimation of life history parameters in such a rapidly breeding taxa.

RESPONSE: While survival was recorded daily, for logistical reasons, fecundity was recorded twice a week (during water changes). Similar protocol has been utilised in many other studies (e.g. Rogalski & Duffy 2020 *Evolution*, Hall & Ebert 2012 *Proceedings B*, Flaherty & Dodson 2005 *Chemosphere*). We have changed the wording to make this clearer as follows:

“Individuals were monitored daily for survival and the number of offspring and clutches produced was counted twice a week at each water change.” (lines 145-146)

REFEREE COMMENT: L158. Please state survival in all treatments. And please include data on timing and brood size in each instar for each treatment, at least in an appendix. Were all offspring born healthy? Did offspring differ in size?

RESPONSE: We have not included survival and deaths in our results section due to the low overall number of deaths across all treatments; however this information, along with timing and brood size of each instar will be freely accessible on our Dryad repository. We were unable to measure offspring size due to logistical constraints.

REFEREE COMMENT: L163. Although not statistically significant, the low size of first clutch at 25°C is noticeable and worth commenting on.

RESPONSE: Due to the large number of traits, variables and interactions in our study, we have chosen to focus only on results with clear statistical support, and the broader pattern overall. As such, we feel it would not be suitable to comment on the non-statistically low size of first clutch at 25°C, particular as it is also only seen in one genotype.

REFEREE COMMENT: Fig. 2. There is too much information in graphs C and D. Either find a way to simplify or delete. I don't think they add a lot anyway.

RESPONSE: We appreciate the reviewer's point, and agree that simpler graphical information is usually better. However, the information provided in graphs C and D is essential for readers to properly interpret graphs A and B, so we have chosen to retain these graphs.

REFEREE 2:

REFEREE COMMENT: This manuscript is a novel contribution to unraveling the complex responses of organisms to multiple potential stressors (e.g., antidepressants and temperature). The experiments and analyses are sound and the results are well supported and of interest. The manuscript was well written and the results were presented very clearly. In general, the manuscript provides a clear description of the need for this research and a good description of the research conducted. These types of experiments are imperative for understanding how multiple stressors affect organisms in nature, where multiple stressors exist.

RESPONSE: We thank the referee for their positive feedback.

REFEREE COMMENT: My only comments for revision are to request that the authors provide a bit more detail about the study organism and its thermal optimum and how the temperature treatment compares to their thermal optimum. The authors indicate that the higher temperatures represent a stressor, but do not provide adequate information in the text to ascertain how sensitive *D. magna* is to a 20 vs. 25C thermal regime.

RESPONSE: Reviewer 1 raised a similar point which we have now addressed through adding the following sentence:

*“The temperature treatments used were 20°C, which is the standard cultivation temperature of *Daphnia* [26, 48-50], and 25°C, which is known to affect “pace-of-life” traits, often leading to significantly faster maturation, earlier offspring release and smaller size at maturity, but at the expense of reduced survival and lifetime fecundity (e.g. [51-53]).”* (lines 90-94)

REFEREE COMMENT: Along the same lines, in the discussion the authors make some statements which I think need to be refined given thermal regimes in lakes, stratification and the potential behavior of *Daphnia*. The statement “Indeed, warmer temperatures appear to negate the effects of fluoxetine on an organism’s life-history, suggesting that the effects of this widespread pharmaceutical will not necessarily be made worse under common scenarios of global change.” And other similar statements in the discussion could be refined because in any given lake or pond where *Daphnia* exist, they will be exposed to a variety of temperatures. *Daphnia* are also known to migrate vertically in the water column as well (thereby experiencing different temperatures).

So, although the point is certainly useful that there is a strong potential for temperature to interact with contaminants like the one studied here, how this plays out in a given ecosystem will be very complex. I would recommend that the authors avoid the word “negate” and stress the interactions of stressors and their findings demonstrate that the

effects of this pollutant are strongly dependent on the temperature during exposure, at least within the rather narrow range that has been examined

RESPONSE: Fair comments. We are now more circumspect in our wording, especially in light of the ecological complexities that exist in nature (whilst keeping within the strict word limit of Proceedings B). In this regard, while we do not have the space to discuss vertical migration in depth, it is worth noting here that this behaviour, including in response to temperature, can still result in heightened fitness cost given temperature-related metabolic demands (See Muller *et al.* 2018 *PloS one*). In any case, taking on board the reviewer's broader points around ecological complexity, we have amended relevant passages of the text now read, as follows:

"A 5°C increase in temperature led to all treatments (i.e. freshwater control and fluoxetine exposures alike) to converge on a common life-history phenotype (Figure 2), suggesting that rising temperatures may potentially reduce the net phenotypic effects of fluoxetine pollution on an individual's life-history in some contexts." (lines 222-226)

*"Regardless of the underlying mechanism, our results suggest that the effects of the pollutant fluoxetine will not necessarily be exacerbated under the rise in temperatures predicted for many scenarios of global change. This demonstrates that while climate change is often predicted to amplify threats to ecosystems, this is not always inevitable (see also [39, 76]), although due to the complex nature of ecosystems, the exact effects are likely to depend on context, such as the type of pollutant, the type of thermal change, and as we discuss below, the genetic background of the exposed individual. In particular, in any lake or pond where *Daphnia* exist, they will likely be exposed to a variety of temperatures, due to spatial and temporal variation in thermal regimes, and their own ability to migrate vertically in the water column [77-79]. In the wild, the potential for temperature change to limit the impact of fluoxetine will depend strongly on this fine-scale variation in temperature and the exposed individual's own thermal preference."* (lines 232-244)

"Indeed, warmer temperatures appear to lessen the effects of fluoxetine on an organism's life-history, suggesting that the effects of this widespread pharmaceutical will not necessarily be made worse under common scenarios of global change." (lines 271-274)